Mitigative effect of sodium alginate on streptozotocin (STZ)-induced diabetic neuropathy through regulation of redox status and miR-146a in the rat sciatic nerve

Mohamed Nema A. 1
Shouran Naeimah M. 1 2
Essawy Amina E. 1
http://orcid.org/0000-0002-0270-703X Abdel-Moneim Ashraf M. 1 3
http://orcid.org/0000-0003-3269-7458 Abdel Salam Sherine 1 3 saimohamed@kfu.edu.sa
1 Department of Zoology, Faculty of Science, Alexandria University , Alexandria , Egypt
2 Department of Zoology, Faculty of Science, Bani Waleed University , Bani Waleed , Libya
3 Department of Biological Sciences, Faculty of Science, King Faisal University , Al-Ahsa , Saudi Arabia
van der Westhuizen Francois
Electronic publication date: 2025 Mar 24
Publication date: 2025
Volume: 13
Electronic Location ID: e19046
Received 2024 Oct 8; Accepted 2025 Feb 3
Copyright: © 2025 Mohamed et al.
Copyright year: 2025
Copyright holder: Mohamed et al.
License: This is an open access article distributed under the terms of the Creative Commons Attribution License, which permits unrestricted use, distribution, reproduction and adaptation in any medium and for any purpose provided that it is properly attributed. For attribution, the original author(s), title, publication source (PeerJ) and either DOI or URL of the article must be cited.
License URL: https://creativecommons.org/licenses/by/4.0/

Keywords: Sodium alginate, Diabetic neuropathy, Oxidative stress, Inflammation, Apoptosis, MiR-146a

Funding: The authors received no funding for this work.

==============================
Diabetic peripheral neuropathy (DPN) is a significant complication of diabetes with limited effective therapeutic options. Sodium alginate (SA), a natural polysaccharide from brown algae, has demonstrated health benefits, however, whether it can treat streptozotocin (STZ)-induced DPN remains unclear. The present experiment aimed to test the preventive role of SA on STZ-induced DPN in rats and explored the possible mechanisms. The DPN rat model was established in rats by intraperitoneal injection of a single dose of 40 mg/kg b.w. STZ, and SA (200 mg/kg b.w./day) was orally administered for 28 days after type 2 diabetes mellitus (T2DM) induction. The obtained findings revealed that STZ significantly increased serum levels of FBG, HOMA-IR, TC, TG, VLDL-C, and LDL-C, while decreased serum insulin, incretin GLP-1, HDL-C, and lipase activity. In the sciatic nerves, STZ significantly increased proinflammatory cytokine levels (IL-1β, IL-6, and TNF-α), caspase-3 (a pro-apoptotic protein), markers of oxidative stress (MDA and NO), and AGEs. In parallel, STZ induced a significant decline in the activities of enzymatic antioxidants, viz., SOD, CAT, and GPx, and non-enzymatic GSH. These changes were accompanied by a low expression of miR-146a in the sciatic nerves of DPN rats. Except for HOMA-IR, SA treatment to STZ injected rats significantly improved these parameters and helped to rescue the neurological morphology of the sciatic nerve fibers. In conclusion, SA mitigated experimental DPN, and this might be due to its ability to suppress hyperglycemic-hyperlipidemic effects, counteract the overactivation of inflammatory molecules, increase miR-146a expression, modulate oxidative dysregulation, and reduce cell apoptosis.

Introduction

Diabetes mellitus (DM) is a metabolic disease that poses a serious threat to public health. In 2019, the prevalence of diabetes was around 9.3% (463 million people), and it is expected to rise to 10.2% (578 million) by 2030. Also, it has been estimated to reach 10.9% (700 million) by 2045 (Saeedi et al., 2019). In DM, long-term hyperglycemia progresses gradually and can lead to a range of complications that affect the kidneys, eyes, nerves, and cardiovascular system (Galicia-Garcia et al., 2020). Diabetic peripheral neuropathy (DPN) is the most prevalent complication in diabetes, affecting over 50% of type 2 diabetic patients (Feldman et al., 2019) and can cause dysfunctional sensory symptoms such as burning, hyperalgesia, allodynia, and dysesthesia (Javed et al., 2020). DPN can also predispose patients to the risk of foot ulcerations and amputations (Selvarajah et al., 2019). Several pathogenic factors are known to be involved in the development of DPN complications, including hyperglycemia-induced oxidative cellular injury, microvascular damage, and altered insulin receptor pathways (Singh, Kishore & Kaur, 2014; Zhu et al., 2024). Moreover, studies have shown that DPN is closely associated with the increase of immune-inflammatory response (Pop-Busui et al., 2016). Recent evidence indicates that miRNAs could represent highly valued therapeutic targets for the treatment of neurological disorders, including peripheral neuropathies (Fan et al., 2020). Feng et al. (2018) reported that the expression level of miR-146a decreases under long-term hyperglycemia, and results in excessive expression of cytokines via NF-kB activation in a rat model with DPN. Recently, metformin, an anti-type 2 DM (anti-T2DM) drug, was found to modulate inflammation in DPN through increased miR-146a expression (Liu et al., 2024). Previous research indicated that miR-146a inhibits cytokines secretion by suppressing expression levels of interleukin 1 receptor-associated kinase 1 (IRAK1) and tumor necrosis factor receptor-associated factor 6 (TRAF6), which are key adaptor molecules in the Toll-like receptors (TLR)/NF-kB signaling pathway (Hassan et al., 2024). It has been also proven that miR-146a can repress inflammation in other non-neurological disease models (Runtsch et al., 2019; Liao, Zheng & Shao, 2023).

Currently, there are few analgesic treatments for DPN, which only provide pain relief, and so it is an urgent need to explore more effective neuroprotective drugs. Removal of free radicals by natural and synthetic antioxidants may counteract oxidative damage and prevents the development of neuropathy. In recent decades, a rapidly growing number of biomaterials with free radical scavenging properties were reported to confer protection against diabetic nerve injuries and reduce the progression of DPN (Naseri et al., 2019; Qi et al., 2020; Saraswat, Sachan & Chandra, 2020; Xie et al., 2020; Gölboyu et al., 2024). Sodium alginate (SA) is a natural polysaccharide product obtained from brown algae, which consists of two uronic acids, namely β-D-mannuronic acid (ManA) and α-L-guluronic acid (GulA) (Sosnik, 2014). It is a non-toxic biopolymer used in a wide range of applications, such as cosmetics, pharmaceutical and food industry (Tønnesen & Karlsen, 2002). Previously published data suggested that alginates exert enormous and vital biological properties, including anti-inflammatory, antioxidant and anti-apoptotic actions against neurodegenerative diseases (Tusi et al., 2011), anticancer drug side-effects (Yamamoto et al., 2013; Samir et al., 2023; Moradi et al., 2024), myocardial reperfusion injury (Guo et al., 2017), and metal toxicity (Gao et al., 2020). Other researchers (Acevedo et al., 2024) showed that alginate oligosaccharides (AOSs) could protect against peroxide-induced oxidative injury in the gastric epithelial cell line by activating the Nrf2 pathway, which is crucial for regulation of cellular redox homeostasis. Furthermore, supplementation of SA from seaweed Sargassum hystrix was able to lower glucose and limit necrosis of pancreatic β-cells in STZ-induced T2DM rats (Gotama, Husni & Ustadi, 2018). However, to date, few in vivo studies have examined the neuroprotective effect of SA (e.g., Saleh, Hamdy & Hassan, 2022) and none of these used the DPN model. Based on this background, the present study aimed to elucidate the molecular and physiological mechanisms by which SA confers protection against inflammation activation, ROS production, and apoptosis in STZ-induced DPN. The findings presented here provide miR-146a as a new target for SA in the treatment of DPN.

Materials and Methods

Chemicals

STZ was supplied from Sigma Chemical Company, Saint Louis, MO, USA. SA was purchased from Alpha Chemika, Mumbai, India (batch number: SA579). All other chemicals were of high purity and analytical grade.

Animals and induction of T2DM

The Alexandria University- Institutional Animal Care and Use Committee (AlexU-IACUC) has approved this study (Ref. No.: AU 04 22 11 26 1 02 dated 26 November 2022). Male Wistar albino rats at 6 weeks old and weighting 180–200 g were procured from an animal breeding facility, Alexandria University, Egypt. The animals were placed separately (two or three rats per cage) and housed in polypropylene cages (length, 45 cm; width, 30 cm; height, 20 cm) containing wood-chip bedding material. Rats were kept at ambient temperature (23 ± 3 °C) with 12-h light/dark cycle and free access to a standard pellet diet and water. They were left to acclimatize for a period of 2 weeks prior to the initiation of treatments. T2DM was induced as previously reported (Aloud et al., 2017). Briefly, rats received a single intraperitoneal (i.p.) injection of 40 mg/kg body weight (b.w.) of STZ freshly dissolved in a 0.1 M/l citrate buffer (pH 4.5). To avoid hypoglycemic shock, these animals were allowed to drink a 20% sterile glucose solution for the first 24 h. Fasting blood glucose (FBG) was assessed three days after STZ injection using a glucometer (Frankenberg, Germany), and rats showing FBG levels > 200 mg/dl were considered to have T2DM (Al-Awar et al., 2016) and were included in subsequent experiments.

Experimental design

Rats were randomly allocated into four groups (n = five in each) as; control: normal (nondiabetic) rats maintained on the standard diet, STZ: STZ-induced diabetic rats without treatment, STZ+SA: STZ-induced diabetic rats were orally given 200 mg/kg b.w./day of SA (dissolved in saline) by gastric gavage, and SA: normal rats were orally given doses of SA (200 mg/kg b.w./day) (Fig. 1). The 200 mg/kg dose of SA (therapeutic dose) has been found to exhibit no toxicity and promising efficacy in preventing neuronal damage following brain injury in rats (Saleh, Hamdy & Hassan, 2022). Rats in the control and SA treated groups received the same volume of vehicle (citrate buffer) in place of STZ and all treatments with SA were performed for 28 days. On the 28th day, all rats were fasted overnight for 12 h and then euthanized by inhaled isoflurane overdose, and maximum efforts were made to reduce and/or limit suffering of the animals.

Figure 1 Experimental timeline showing treatment schedule for STZ and SA.

Collection of tissue samples

Blood was withdrawn from the rats by intracardiac puncture and serum was separated for biochemical tests. The sciatic nerves were isolated from the two hindlimbs as described earlier (Saraswat, Sachan & Chandra, 2020). The right sciatic nerve of each rat was cleaned from blood, and then a homogenate was prepared with 2 ml phosphate saline (50 mM potassium phosphate pH 7.5, 1 mM EDTA). The homogenate was centrifuged at 5,000 × g for 10 min at 4 °C. The supernatant was directly stored at −20 °C until the assay. The left sciatic nerve was quickly frozen at −80 °C until qRT-PCR testing, and representative pieces of tissues were fixed in a convenient fixative for histopathological and ultrastructural examinations.

Serum biochemical assays

FBG levels were measured with Biodiagnostic kit, Giza, Egypt (Cat. Number: GL 13 20) according to glucose-oxidase-peroxidase method. Serum insulin concentration was determined by an ELISA technique (Cat. Number: ERINS; Thermo Fisher Scientific, Waltham, MA, USA) with the intra-assay coefficient of variation percent (CV%) < 10% and the inter-assay CV% < 12%. Insulin resistance (IR) was estimated with the formula: homeostatic model assessment index (HOMA)-IR = [FBG (mg/dl) × fasting insulin (mU/ml)]/405 (Matthews et al., 1985). Incretin GLP-1 level was measured by ELISA using a kit supplied by Shibayagi Co., Ltd., Japan (Cat. Number: RSHAKMGP-011R) with the intra-assay and inter-assay precision CV% < 5%. TC and TG was quantified using standard enzymatic methods (Allain et al., 1974; Fossati & Prencipe, 1982). HDL-C was determined by phosphotungstic acid/magnesium ions precipitation procedures (Lopes-Virella et al., 1977). VLDL-C and LDL-C were calculated from the equations of Friedewald as follows: VLDL-C = TG/5 and LDL-C = TC – [HDL-C + VLDL-C] (Friedewald, Levy & Fredrickson, 1972). Lipase activity was assayed as per the manufacturer’s instructions using a commercial kit (Cat. Number: LS-K298-100; LifeSpan BioSciences, Inc., Seattle, WA, USA).

Determination of inflammatory markers and caspase-3 in the sciatic nerve

Specific ELISA kits were used to determine proinflammatory cytokines IL-1β (Cat. Number: E0119Ra), IL-6 (Cat. Number: E0135Ra), and TNF-α (Cat. Number: E0764Ra) as provided by the corresponding manufacturer’s manual (Bioassay Technology Laboratory BT Lab, Shanghai, China). The apoptosis-related marker (caspase-3) was estimated by ELISA using a rat caspase-3 kit (Cat. Number: CSB-E08857r; CUSABIO, Wuhan, Hubei, China). The intra-assay and inter-assay CV% were 4.3–4.8% and <10% for IL-1β, 4.5–5.7% and <10% for IL-6, 4.6–5.7% and <10% for TNF-α, and <8% and <10% for caspase-3, respectively, according to the corresponding manufacturer’s protocol.

Measurement of oxidative stress indices and AGEs in the sciatic nerve

The level of MDA, a product of lipid peroxidation, was determined using the reactive thiobarbituric acid substances (TBARs) assay (Ohkawa, Ohishi & Yagi, 1979). NO, a marker of RNS, was measured by colorimetry as its metabolic nitrite following the Griess reaction method (Montgomery & Dymock, 1961). For antioxidant enzymes, the methods outlined by Nishikimi, Appaji Rao & Yagi (1972), Aebi (1984), and Paglia & Valentine (1967) were used to estimate the activities of SOD, CAT, and GPx, respectively. GSH content was assessed by employing the Beutler’s technique (Beutler, Duron & Kelly, 1963). AGEs were determined using an ELISA kit (Cat. Number: CEB353Ge; Cloud-Clone Crop, Wuhan, China). The intra-assay CV% for this test, as reported by the manufacturer, was <10%, and the inter-assay CV% was <12%.

Quantitative RT-PCR analysis for miR-146a

Total RNA was extracted from frozen nerve tissues using the GENEzol TriRNA Pure kit (GZX050; Geneaid Biotech Ltd, New Taipei, Taiwan) as per the manufacturer’s guidelines. For each sample, sciatic nerve tissue was transferred to RNAse free 1.5 ml centrifuge tube. To avoid DNA contamination, 700 µl of GENEzol reagent was added in each tube followed by tissue grinding. The purity, quality and quantification of extracted RNA was assessed using Nandrop 2000 (Thermo Fisher Scientific, Waltham, MA, USA). To ensure the optimum stability of extracted RNA, RNA wash and elution using RNAse free water were finally performed. The obtained A260/280 ratio was around 1.7 and 2.0. The COSMO cDNA PLUS synthesis kit (Willowfort, WF10205006, Birmingham, UK) was used to reverse transcribe RNA into cDNA. According to manufacturer’s guidelines, 400 ng of the purified RNA samples, 4 µl of cDNA reaction buffer, 1 µl of RT enzyme mix were used. Using advanced primus 25 thermocycler PCR (Peqlab, Erlangen, Germany), the following conditions were optimized: for primer annealing (25 °C, 5 min), for extension (55 °C, 15 min), and for inactivation (85 °C, 5 min). cDNAs were stored at −20 °C for further usage. No multiplex PCR was used in this study. qRT-PCR was performed using the HERAPLUS SYBER Green qPCR kit (WF1030800X; Willowfort, Birmingham, UK). The reactions were completed on a BIO-RAD CFX real-time system (Bio-Rad, Kaki Bukit, Singapore). All PCR reactions were performed under standard PCR conditions using nuclease free eppendorfs. Briefly, a rection mixture of 20 µl including 2 µl of cDNA sample, 10 µl of HeraPlus SYBER® Green Master Mix, and 1 µl (200 nM) of each primer were completed to 20 µl with nuclease free water. qRT-PCR was carried out according to the following stages: stage 1 (95 °C, 2 min) followed by 40 cycles of stage 2 (2.1 (95 °C, 10 s) and stage 2.2 (60 °C 30 s). The Ct values were acquired and the relative quantity of miR-146a was normalized to the internal control U6 snRNA. The expression level of miR-146a was calculated according to the 2−ΔΔCt formula (Livak & Schmittgen, 2001). The primers sequences and miRBase accession numbers were provided by Vivantis Technologies (Selangor, Malaysia) (Table S1).

Light and transmission electron microscopy

Sciatic nerve tissues (blocks of 2 mm3 each) were fixed in 4F1G (4% formaldehyde and 1% glutaraldehyde) solution in phosphate buffer (pH 7.2) for 3 h at 4 °C, and then they were immersed in 2% osmium tetroxide at 4 °C for 2 h (postfixation), washed in buffer, dehydrated through ascending ethanol series, and finally embedded in epon-araldite resin mixture. Semithin sections (1 μm thick) were stained with 1% toluidine blue in a buffer of 1% sodium tetraborate and observed using a light microscope (Olympus BX51; Olympus, Tokyo, Japan). Next, ultrathin sections (50 nm thick) of selected areas were placed on copper grids and contrast stained with a solution of uranyl acetate and lead citrate (Reynolds, 1963), thereafter they were examined at 80 kV acceleration voltage in a JEM-1400 TEM (JEOL Ltd., Tokyo, Japan).

Statistical analysis

The data was processed with the SPSS software package version 21.0 (SPSS Inc, Chicago, IL, USA) and presented as means and standard errors (SEs). One-way analysis of variance (or one-way ANOVA) was undertaken to compare multiple groups, followed by LSD post-hoc test for pairwise comparisons, when appropriate. In addition, statistical correlations between selected indexes were conducted using the Pearson’s correlation analysis and linear regression, where values of the Pearson correlation coefficient (r) less than (0.19), (0.2−0.39), (0.4−0.59), (0.6−0.79), and more than (0.79) was considered a very weak, weak, moderate, strong, and very strong correlation, respectively (Senousy et al., 2022). Finally, principle component multivariate analysis (PCA) was performed using Minitab 17.0 to visualize clustering among the treatment groups. The level of significance (p-value) was adjusted at p < 0.05.

Results

Serum biochemical parameters

One-way ANOVA revealed significant variations among study groups in FBG (F(3. 16) = 93.977, p < 0.001), HOMA-IR (F(3. 16) = 6.766, p = 0.004), insulin levels (F(3. 16) = 79.710, p < 0.001), and incretin GLP-1 (F(3. 16) = 123.095, p < 0.001). Post-hoc comparisons indicated that injection of STZ caused a significant increase in FBG (Fig. 2A) and HOMA-IR (Fig. 2B), and a significant decrease in blood insulin (Fig. 2C) and incretin GLP-1 (Fig. 2D) compared to the control (LSD test: FBG (p < 0.001), HOMA-IR (p = 029), insulin (p < 0.001), and incretin GLP-1 (p < 0.001)). For the STZ+SA group, the analysis showed that all of the aforementioned perturbations (except for HOMA-IR) were significantly (p < 0.001) improved, although not reaching the normal levels observed in the control animals.

Figure 2 SA induces alleviation of glycemic markers in the diabetic (STZ-treated) rats.

Treatment of rats with SA (200 mg/kg b.w.) for 28 days significantly increases FBS (A) without changing HOMA-IR (B), and prevents the decrease in insulin levels (C) and incretin GLP-1 (D) in diabetic model. Data are expressed as means ± SEs and analyzed by one-way ANOVA followed by LSD post-hoc test (n = five animals per group). Individual data points are shown as yellow dots. ns p > 0.05 (non-significant), *p < 0.05 compared to the control, #p < 0.05 compared to the diabetic (STZ) group.

Regarding blood lipids, multiple group comparisons using one-way ANOVA demonstrated significant differences in TC (F(3. 16) = 37.173, p < 0.001), TG (F(3. 16) = 30.120, p < 0.001), VLDL-C (F(3. 16) = 30.120, p < 0.001), LDL-C (F(3. 16) = 54.896, p < 0.001), HDL-C (F(3. 16) = 37.543, p < 0.001), and lipase activity (F(3. 16) = 65.247, p < 0.001). LSD post-hoc analysis showed that there was a significant (p < 0.001) increase in TC (Fig. 3A), TG (Fig. 3B), VLDL-C (Fig. 3C), and LDL-C (Fig. 3D) in STZ-induced diabetic rats compared to their corresponding controls, whereas HDL-C (Fig. 3E), and lipase (Fig. 3F) were significantly (p < 0.001) reduced. On the contrary, levels of TC, TG, VLDL-C, and LDL-C significantly (p < 0.001) decreased in the therapy group (STZ+SA), and HDL-C and lipase enzyme significantly (p = 0.001 and p < 0.001) augmented compared to STZ model but not the normal control range.

Figure 3 SA attenuates dyslipidemia in the diabetic (STZ-treated) rats.

Treatment of rats with SA (200 mg/kg b.w.) for 28 days significantly reduces TC (A), TG (B), VLDL-C (C), and LDL-C (D), while increases HDL-C (E) and lipase activity (F) in diabetic model. Data are expressed as means ± SEs and analyzed by one-way ANOVA followed by LSD post-hoc test (n = five animals per group). Individual data points are shown as yellow dots. ns p > 0.05 (non-significant), *p < 0.05 compared to the control, #p < 0.05 compared to the diabetic (STZ) group.

Proinflammatory cytokines (IL-1β, IL-6, and TNF-α) and pro-apoptotic caspase-3 in the sciatic nerve

Results of one-way ANOVA showed significant discrepancy in levels of IL-1β (F(3. 16) = 148.976, p < 0.001), IL-6 (F(3. 16) = 105.264, p < 0.001), and TNF-α (F(3. 16) = 114.739, p < 0.001) among the studies groups. LSD post-hoc test confirmed that IL-1β (Fig. 4A), IL-6 (Fig. 4B), and TNF-α (Fig. 4C) were significantly (p < 0.001) increased in the sciatic nerves of STZ-induced diabetic rats compared to controls. This inflammatory response was significantly (p < 0.001) reduced after SA intervention (i.e., in the STZ+SA-treated rats) compared to STZ-treated rats, but normalcy was not achieved. Further analysis with one-way ANOVA revealed that the levels of caspase-3 were also altered (F(3. 16) = 69.692, p < 0.001). LSD post-hoc test results showed that apoptosis was significantly (p < 0.001) potentiated in the diabetic group via elevated levels of caspase-3 compared to the control (Fig. 4D). ELISA analysis demonstrated that SA significantly (p < 0.001) downregulated caspase-3 in the STZ+SA-treated rats compared to the diabetic control (i.e., STZ only), although failed to return to normal.

Figure 4 SA downregulates sciatic nerve proinflammatory and apoptotic markers in the diabetic (STZ-treated) rats.

Treatment of rats with SA (200 mg/kg b.w.) for 28 days significantly decreases IL-1ß (A), IL-6 (B), and TNF-α (C), and caspase-3 levels (D) in diabetic model. Data are expressed as means ± SEs and analyzed by one-way ANOVA followed by LSD post-hoc test (n = five animals per group). Individual data points are shown as yellow dots. ns p > 0.05 (non-significant), *p < 0.05 compared to the control, #p < 0.05 compared to the diabetic (STZ) group.

Oxidative stress and levels of AGEs in the sciatic nerve

One-way ANOVA indicated significant effects of treatment groups on MDA (F(3. 16) = 84.069, p < 0.001) and NO (F(3. 16) = 57.825, p < 0.001), CAT (F(3. 16) = 55.555, p < 0.001), SOD (F(3. 16) = 68.900, p < 0.001), GPx (F(3. 16) = 62.116, p < 0.001), GSH (F(3. 16) = 25.020, p < 0.001), and AGEs (F(3. 16) = 79.221, p < 0.001). In LSD post-hoc comparisons, the levels of MDA (Fig. 5A) and NO (Fig. 5B) were significantly (p < 0.001) increased in the STZ-treated rats by +249.58% and +136.26%, while CAT (Fig. 5C), SOD (Fig. 5D), GPx (Fig. 5E), and GSH (Fig. 5F) were significantly (p < 0.001) diminished by –63.60%, –59.71%, –62.91%, and –71.04%, respectively, compared to the control. Furthermore, STZ-induced diabetic rats had significantly (p < 0.001) increased AGEs (Fig. 5G) in their sciatic nerve tissues by +86.38% compared to the control. In contrast, diabetic rats treated with SA showed significant (p < 0.001) decrease in the levels of MDA (–38.97%), NO (–31.46%), and AGEs (–26.95%) compared to the STZ diabetic group. Moreover, CAT, SOD, GPx, and GSH were significantly increased in STZ+SA-treated rats by (+80.75%), (+78.85%), and (+75.91%), and (+88.44%), respectively, compared to STZ-alone rats (LSD post-hoc between STZ+SA and STZ: CAT (p < 0.001), SOD (p < 0.001), GPx (p < 0.001), and GSH (p = 0.013)). However, in all these parameters, significant differences were still shown between STZ+SA vs the control.

Figure 5 SA modulates sciatic nerve oxidant-antioxidant parameters and AGEs in the diabetic (STZ-treated) rats.

Treatment of rats with SA (200 mg/kg b.w.) for 28 days significantly decreases the production of MDA (A) and NO (B), increases the levels of CAT (C), SOD (D), GPx (E), and GSH (F), and reverses the increase in AGEs (G) in diabetic model. Data are expressed as means ± SEs and analyzed by one-way ANOVA followed by LSD post-hoc test (n = five animals per group). Individual data points are shown as yellow dots. ns p > 0.05 (non-significant), *p < 0.05 compared to the control, #p < 0.05 compared to the diabetic (STZ) group.

Expression of MiR-146a in the sciatic nerve using qRT-PCR analysis, and the correlations between MiR-146a and tissue biomarkers

One-way ANOVA depicted an overall statistically significant change in miR-146a (F(3. 16) = 7.203, p = 0.003). LSD post-hoc revealed that miR-146a expression was significantly (p = 0.005) downregulated in the sciatic nerves of the rats in STZ group compared to control, while it was significantly (p = 0.001) upregulated and restored to normal levels in the STZ+SA group (Fig. 6A). A significant negative correlation was detected between miR-146a relative expression and IL-1β (Fig. 6B), IL-6 (Fig. 6C), TNF-α (Fig. 6D), and caspase-3 (Fig. 6E) with r values of −0.701, −0.629, −0.719, and −0.605, respectively. These data identify multiple strong associations between miR-146a impairment and the promotion of proinflammatory cytokines and caspase-3 levels.

Figure 6 The effect of SA on the expression level of sciatic nerve miR-146a in the diabetic (STZ-treated) rats.

Treatment of rats with SA (200 mg/kg b.w.) for 28 days significantly upregulates miR-146a (A) in diabetic model. Data are expressed as means ± SEs and analyzed by one-way ANOVA followed by LSD post-hoc test (n = five animals per group). Individual data points are shown as yellow dots. ns p > 0.05 (non-significant), *p < 0.05 compared to the control, #p < 0.05 compared to the diabetic (STZ) group. Correlation between miR-146a expression with the levels of IL-1β (B), IL-6 (C), TNF-α (D), and caspase-3 (E) in the sciatic nerve. Correlation coefficient (r) and p-value were determined using the Pearson’s correlation analysis, and solid line represents linear regression fit.

Furthermore, the statistical results demonstrated that there were significant moderate negative correlations between miR-146a expression level and MDA (r = −0.599, Fig. 7A) and NO (r = −0.545, Fig. 7B), while there were significant moderate-to-strong positive correlations with SOD (r = 0.587; Fig. 7C), CAT (r = 0.609; Fig. 7D), GPx (r = 0.588, Fig. 7E), and GSH levels (r = −0.543, Fig. 7F). In addition, miR-146a showed a significant strong negative correlation with AGEs (r = −0.623, Fig. 7G). Patterns of these correlations suggest that STZ-induced miR-146a inhibition in the sciatic nerve was related to the decrease of antioxidant activities and elevation of oxidant content.

Figure 7 Correlation between miR-146a expression with the levels of MDA (A), NO (B), SOD (C), CAT (D), GPx (E), GSH (F), and AGEs (G) in the sciatic nerve.

Correlation coefficient (r) and p-value were determined using the Pearson’s correlation analysis, and solid line represents linear regression fit.

PCA findings for biochemical and molecular variables

Scree plot (Fig. 8A) depicts the Eigenvalues and the principle components (i.e., Eigenvectors) after the elaboration of the whole dataset. The first principle component is responsible for more than 86.3% of the total data variability, and the second principle component contributed 4.5% of the total variance. Other principle components with much lower Eigenvalues were discarded. In the PCA scatter plot (Fig. 8B), it can be seen that there was a clear trend of separation between the control-SA cluster and the STZ (or the model) cluster. What is more, better therapeutic effect of SA was confirmed by discrimination between the STZ and STZ+SA clusters. Furthermore, STZ+SA still could be distinguished from the control-SA aggregated cluster.

Figure 8 Scree plot (A) depicting the eigenvalues of the factors extracted by factor analysis using biochemical and molecular indices as variables, and PCA score plot (B) for PC1 vs. PC2 showing a separation of control-SA cluster, diabetic (STZ-treated) and STZ+SA groups.

Morphological evaluation of nerve changes

TEM investigations showed normal architecture of myelin in the control and SA groups, but, in the STZ group, there was an increase in the number of nerve fibers with abnormal myelin, such as focal infoldings, intramyelinic edema and distortion of myelin lamellae (demyelination) with accompanying axonal atrophy. Treatment with SA decreased the intensity of nerve lesions but did not prevent abnormal cases (Fig. 9). Semithin histologic images, shown in Fig. S1, confirmed the STZ-induced nerve damage and the partial recovery upon SA treatment.

Figure 9 Effect of SA on ultrastructure of sciatic nerves in the diabetic (STZ-treated) rats under transmission electron microscopy.

STZ representative image exhibits destroyed myelin with vacuoli-zation and lamellar separation in the myelin fibers and shrunken axons. While, in the STZ+SA, the structure of the myelin sheath is partially preserved compared to STZ. No obvious abnormalities in myelination are seen in the control and SA. Note axons (Ax), myelin (M), demyelinating lesions (arrows), and atrophied axon (asterisk) with altered mitochondria. Scale bars = 1 µm (control panel), 2 µm (STZ and SA panels), and 5 µm (STZ+SA panel).

Discussion

This study, to our knowledge, is the first to focus on the possible neuroprotective efficacy of SA against STZ-induced T2DM and DPN in rats. The use of low dose of STZ (e.g., 40 mg/kg b.w.) was shown to result in partial destruction of the pancreatic islet β-cells and mild insulin deficiency. Such effects sufficiently lead to develop non-insulin-dependent diabetogenic state, more closely resembling that of T2DM (Toma et al., 2015; Aloud et al., 2017). This model is also used to induce diabetic pain and diabetic nerve complications (Nonaka, Akiyama & Une, 2024). High blood glucose and insulin deficits in STZ-induced diabetic rats contribute to decreased nerve growth factor (NGF) in Schwann cells (Ahmad et al., 2022) and the existence of aberrant fibers in the sciatic nerve with axonal degeneration and myelin breakdown that may inhibit nerve activity in DPN (Zangiabadi et al., 2014). In our data, treatment of T2DM rats with 200 mg/kg SA corrected values of FBG, serum insulin level, and lipid profile parameters compared to the STZ-only treated group. Further, SA elevated circulating levels of incretin GLP-1, which, according to the literature, provides a stimulus for insulin gene transcription and regulates β cell turnover with beneficial effects on β cell mass (Perfetti et al., 2000). In harmony with our study, the hypoglycemic and antidyslipidemic effects of SA have been ascertained in high fat diet-induced obese mice (Wang et al., 2018; Qiang et al., 2022). In this regard, in C2C12 myoblasts, the alginate-derived oligomannuronate (OM)-chromium (III) complex was reported to have a high antidiabetic potential through upregulation of insulin receptor and glucose transporter-4 expression due to stimulation of insulin (PI3K/Akt) and AMPK signaling pathways (Hao et al., 2011). In addition, both OM and OM-chromium (III) complex were reported to promote lipid metabolism by enhancing fatty acid β-oxidation, while augmenting triacylglyceride lipase protein expression and lipolysis in the 3T3-L1 adipocytes (Hao et al., 2015). Furthermore, Liu et al. (2020) indicated that OM prevents mitochondrial dysfunction and cell apoptosis in pancreatic β cell lines by suppressing the JNK pathway.

Hyperglycemia and dyslipidemia result in release of abundant amounts of ROS due to oxidative stress (González et al., 2023), which can cause elevated immunological stimulation and nerve demyelination, and ultimately lead to DPN and pain symptoms (Chong, Menkes & Souayah, 2024). Persistent hyperglycemia-induced glycation of proteins, lipids, and nucleic acid (i.e., AGEs) potentially exacerbate oxidative stress and inflammatory processes (Khalid, Petroianu & Adem, 2022). The extensive deposition of AGEs also triggers apoptotic cell death in Schwann cells, which can be followed by demyelination of nerve fibers (Wang et al., 2023). In our work, we found that SA reduced axon demylenation in sciatic nerves of diabetic rats as revealed by histological and ultrastructural examinations. Meanwhile, after the SA treatment, the diabetic animals exhibited a significantly decreased proinflammatory factors (like IL-1β, IL-6, and TNF-α), caspase-3, oxidative stress levels, and AGEs, while enhanced the antioxidant defense. The beneficial role of SA in DPN rats may be explained by the presence of its several antioxidant scavengers, including fucoidans (Atashrazm et al., 2015). Previous findings showed that alginate attenuated oxidative stress in D-Galactose-induced kidney aging through increasing antioxidative SOD and CAT activity and lowering MDA levels (Pan et al., 2021). Furthermore, in the present study, the observed reduction in proinflammatory cytokine release induced by SA was accompanied by upregulation of miR-146a expression. Research on DPN in mice has shown that hyperglycemia causes downregulation of miR-146a expression, which is inversely related to inflammatory IRAK1 and TRAF6 protein levels in dorsal root ganglion (DRG) neurons (Wang et al., 2014). MiR-146a is an endogenous suppressor of IRAK1 and TRAF6, which were shown to mediate the activation of inflammatory NF-κB pathway (Hou et al., 2009). Studies have also shown that miR-146a directly reduces production of inflammatory cytokines in immune cells, particularly macrophages (Li et al., 2013). Saba, Sorensen & Booth (2014) showed that miR-146a influences the innate immune system’s activity and modulates TLR signaling and cytokine response. Furthermore, the increase in miR-146a-5p expression and blocking NF-κB signaling pathways may enhance endothelial function and prevent the development of diabetic complications (Kamali et al., 2016). This notion is also supported by the work of Ye & Steinle (2016) showing that miR-146a is a key player in relieving hyperglycemia-induced inflammation through reduction of TLR4/NF-κB and TNFα signaling cascade. Of note, via interacting with Keap1, overexpression of NF-κB may suppress the Nrf2-antioxidant responsive elements (ARE) pathway (Yu et al., 2011) and thus aggravate oxidative stress burden. On the other hand, another study showed that serum and tissue levels of miR-146a-5p can regulate Nrf2 and NF-κB expressions involved in oxidative stress status and inflammation during the pathogenesis of T2DM (Rasoulinejad, Akbari & Nasiri, 2021). More recently, Zhao et al. (2022) indicated that SA might inhibit NF-κB activity, and this could contribute to lessen the severity of the inflammatory response and oxidative injury. Other antioxidants, including quercetin-3-O-β-galactoside, quercetin, rutin, quercetin-3-O-neohesperdoside and isoquercitrin, showed upregulation of miR-146a expression and downregulation of NF-κB expression in diabetic and obese rats (Abdelhameed et al., 2021). In the literature, there are no studies that examined the in vivo effect of SA on miR-146a expression in the inflammation-driven DPN. Nevertheless, the data in this study offered initial evidence of SA ability to upregulate miR-146a expression, which may inhibit the NF-κB signaling pathway, and detailed research is required to validate this concept and explore additional putative targets of miR-146a in DPN progression. In fact, ManA and GulA, the main residues of AOSs, show excellent immunomodulatory activity and can be used as anti-inflammatory agents (Xing et al., 2020), which is linked to the regulation of TLR4 signaling pathway. In an in vitro study, ManA was reported to exert a suppressor effect on TLR-signaling function and decrease lipoteichoic acid- and lipopolysaccharide-activated production of proinflammatory IL-6 and TNF-α in HEK293 cells (Aletaha et al., 2017). Other studies highlighted the positive therapeutic effects of ManA in animal models of neuroinflammation such as Alzheimer’s disease (Athari Nik Azm et al., 2016) and Epilepsy (Kamali et al., 2020). For GulA, the in vitro study of Hajivalili et al. (2016) indicated that it could reduce the IRAK1 and TRAF6 signaling molecules without altering the expression of miR-146a. It was also reported that GulA plays important role in decreasing the severity of diabetes-induced inflammatory changes in rats (Mortazavi-Jahromi et al., 2020).

In addition to its anti-inflammatory function, miR-146a mimics could inhibit neuronal apoptosis and directly prompt oligodendrocyte progenitor cell differentiation, thereby contributing to increase axonal myelination (Liu et al., 2017). Overexpression of miR-146a-5p inhibits the expressions of transforming growth factor-β (TGF-β) (Min et al., 2017), which is known to disrupt myelination via induction of proapoptotic proteins in experimental DPN (Anjaneyulu et al., 2008). Our data of the lowered caspase-3 levels after SA administration highlights its antiapoptotic activity and this is consistent to some degree with the previous report of Wan et al. (2018) showing that alginate has the ability to inhibit enterocyte death, through reducing apoptosis via mitochondrial-dependent pathway.

Despite providing some valuable insights into the antioxidant/anti-inflammatory effects of SA in experimental model of DPN, this study has limitations, and it should not be linked to clinical practice. First, the sample size in each group was relatively small, and larger samples are needed to verify our findings. Second, there is no reference drug group to compare the efficacy of SA neuroprotection. Third, although we used a literature-based model, the results lack information on the extent of behavioral recovery from pain sensitivity and motor deficits in diabetic rats after SA treatment. Fourth, to provide a more comprehensive understanding of the mechanism of action of SA, future studies should incorporate a broader spectrum of inflammatory and apoptotic proteins using immunohistochemistry and Western blot.

Conclusions

The findings of our study showed that SA not only improves the antioxidant defense system, neuroinflammation, and sciatic nerve damage in STZ-induced diabetic rats, but also exerts hypoglycemic, antidyslipidemic, and anti-apoptotic effects (Fig. 10). Our research also demonstrates that miR-146a is involved in the sciatic nerve inflammation induced by STZ. We propose that peripheral nerve inflammation in diabetic states may be explained by downregulation of miR-146a. Furthermore, SA abolishes the inflammatory response by upregulating miR-146a. The regulation of miR-146a could be, at least in part, one of the mechanisms underlying the neuroprotective effects of SA. These conclusions were further supported by PCA, as the STZ+SA was apparently discriminated from the STZ (diabetic) model. However, evidence in literature is scare to support the use of SA treatment as a stand-alone or adjuvant therapy for T2DM and associated disorders such as DPN, and further preclinical investigations should focus to unravel the complex molecular interplay between miR-146a, its target genes, and other factors in inflammation-mediated DPN.

Figure 10 Schematic representation of the proposed mechanism of action by which SA protects against peripheral neuropathy in the diabetic (STZ-treated) rats.

Supplemental Information

Supplemental Information 1 Raw data.

Supplemental Information 2 qRT-PCR analysis.

Supplemental Information 3 MIQE checklist.

Supplemental Information 4 The ARRIVE guidelines 2.0 Author Checklist.

Supplemental Information 5 Primers of qRT-PCR.

Supplemental Information 6 Representative photomicrographs of sciatic nerve from different groups, i.e., Ctrl: control, STZ: streptozotocin (diabetic), STZ+SA: streptozotocin+sodium alginate, and SA: sodium alginate).

Treatment of rats with SA (200 mg/kg b.w.) for 28 days results in reduction of nerve lesions (arrowheads) in diabetic model. Toluidine blue stained sections. Scale bar: 20 µm.

List of abbreviations

AGEs advanced glycation end-products

AOSs alginate oligosaccharides

CAT catalase

GLP-1 glucagon-like peptide-1

GPx glutathione peroxidase

GSH reduced glutathione

GulA α-L-guluronic acid

HDL-C high-density lipoprotein-cholesterol

HOMA-IR homeostatic model assessment for insulin resistance

IL-1ß interleukin-1beta

IL-6 interleukin-6

IRAK1 interleukin 1 receptor-associated kinase 1

LDL-C low-density lipoprotein-cholesterol

ManA β-D-mannuronic acid

MDA malondialdehyde

NF-κB nuclear factor kappa B

NO nitric oxide

OM Oligomannuronate

PCA principle component analysis

RNS reactive nitrogen species

ROS reactive oxygen species

SA sodium alginate

SOD superoxide dismutase

STZ streptozotocin

T2DM type 2 diabetes mellitus

TC total cholesterol

TG triglyceride

TNF-α tumor necrosis factor-alpha

TLR Toll-like receptors

TRAF6 tumor necrosis factor receptor-associated factor 6

VLDL-C very low-density lipoprotein-cholesterol

Additional Information and Declarations

Competing Interests

The authors declare that they have no competing interests.

Author Contributions

Nema A. Mohamed conceived and designed the experiments, analyzed the data, authored or reviewed drafts of the article, and approved the final draft.

Naeimah M. Shouran performed the experiments, analyzed the data, prepared figures and/or tables, authored or reviewed drafts of the article, and approved the final draft.

Amina E. Essawy conceived and designed the experiments, analyzed the data, authored or reviewed drafts of the article, and approved the final draft.

Ashraf M. Abdel-Moneim analyzed the data, prepared figures and/or tables, authored or reviewed drafts of the article, and approved the final draft.

Sherine Abdel Salam conceived and designed the experiments, performed the experiments, analyzed the data, prepared figures and/or tables, authored or reviewed drafts of the article, and approved the final draft.

Animal Ethics

The following information was supplied relating to ethical approvals (i.e., approving body and any reference numbers):

All procedures using animals in this study were approved by Alexandria University- Institutional Animal Care and Use Committee (AlexU-IACUC) (Ref. No.: AU 04 22 11 26 1 02 dated 26 November 2022).

Data Availability

The following information was supplied regarding data availability:

The raw data are available in the Supplemental Files.

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
