# Peer review of "Mitigative effect of sodium alginate on streptozotocin (STZ)-induced diabetic neuropathy through regulation of redox status and miR-146a in the rat sciatic nerve"

_PeerJ, doi:10.7717/peerj.19046_

## Round 0.1 · original submission · Major Revisions

Four reviews were received, all of which have some minor as well as more substantive comments and suggestions. Please respond to all of these in your next submission.

Reviewer 1 ·

Basic reporting

Mitigative effect of sodium alginate on streptozotocin (STZ)-induced diabetic neuropathy through regulation of redox status and miR-146a in the rat sciatic nerve

The manuscript should be revised for linguistic, grammatical and style errors.
The whole manuscript should be revised for proper use of abbreviations.
The paragraphs should not be started with abbreviations.

Experimental design

The experiment is well designed
Why the nerve didn't stain with HE in different research groups.
Immunostaining is applicable in diabetic peripheral nerve complications to stain inflammatory and apoptotic markers

Validity of the findings

Diabetic complications should be discussed in depth in the introduction and discussion sections.

Additional comments

The limitations of the study should be mentioned clearly.

Reviewer 2 ·

Basic reporting

The study entitled “Mitigative effect of sodium alginate on streptozotocin (STZ)-induced diabetic neuropathy through regulation of redox status and miR-146a in the rat sciatic nerve” is a very interesting and novel work. The authors have shown that several biomaterials, i.e., sodium alginate can also able to treat diabetic peripheral neuropathic condition through regulation of redox status and miR-146a in the STZ-induced rat sciatic nerve. This study further proves that the biomaterials alone can also be utilized as potent therapeutic moiety in the context of various disease apart from drug delivery carrier. The figures illustrated in the manuscript is well analyzed and depicted.

Experimental design

However, the authors can address the following comments or incorporate the suggestions, which will improvise the technical aspect of the manuscript along with reproducibility and readability.
1. The authors suggested to discuss the mechanistic role of STZ to create or induce diabetic peripheral neuropathic condition in rats. This point needs to address for general readers as DPN is a secondary condition or associated disease with diabetes mellitus.
2. The authors suggested to discuss the main or critical challenges of existing therapeutics at molecular level in the context of DPN.
3. The authors suggested to justify how the animal model is appropriate by inducing STZ as there are no specific parameters have been investigated by the authors to identify the DPN condition in rats.
4. The authors suggested to provide the reference or justification to select the dose of SA in the animal model. If the SA didn’t not exhibit its efficacy in DPN model till date, then, how the authors have selected 200 mg/kg b.w once daily dose.
5. The authors suggested to justify how SA was dissolved for the administration purpose.
6. Did you check the toxicity level of SA in other organs or can you please provide the justifications if not checked?
7. Did you check the level of TGF-b in the context of DPN condition.
8. The authors suggested to provide the histopathological slide images in the revised manuscript.
9. Can you please provide the images of pancreas in treatment and control groups, if available.

Validity of the findings

The study entitled is meaningful to replicate and provide valuable insights of utilization of biomaterials in the context of DPN

·

Basic reporting

The manuscript, titled "Mitigative effect of sodium alginate on streptozotocin (STZ)-induced diabetic neuropathy through regulation of redox status and miR-146a in the rat sciatic nerve, " presents well-organized language. However, there are areas for improvement:

Language and Grammar: While generally clear, there are repetitive phrases and occasional awkward sentence constructions, particularly in the results section. For instance, repeated mentions of "STZ+SA significantly improved parameters" can be condensed.
Context and Background: The introduction provides a solid overview of diabetic neuropathy and highlights the novelty of sodium alginate (SA) as a therapeutic agent. However, the literature review could be expanded to include comparative studies with other antioxidants or polysaccharides to better frame SA's significance.
Figures and Tables: The data are well-presented, but some figures lack sufficient labeling (e.g., dose-response relationships for SA are not graphically illustrated). Additionally, the figures are not visually engaging, and their legends require more detail for clarity.

Experimental design

Experimental Design
Research Question: The study aims to address a meaningful gap in DPN treatment by exploring SA’s neuroprotective effects. However, the rationale for the chosen SA dosage (200 mg/kg/day) requires more detailed justification based on previous preclinical studies or pharmacokinetic profiles.
Methodology:
Overall, the methods are well-described, but the mechanism of SA's effects on miR-146a regulation requires further elaboration. As mentioned in the discussion, incorporating immunohistochemistry or Western blot techniques would strengthen the results.
The animal model is appropriate, but additional controls (e.g., comparison to a standard treatment for DPN) would enhance the study's robustness.
General Comments
Strengths:
The study explores a novel, cost-effective, and natural therapeutic approach to DPN.
Strong emphasis on oxidative stress, inflammation, and miR-146a pathways.
Weaknesses:
Lack of functional/behavioral data.
Absence of comparative analysis with established treatments for DPN.

Validity of the findings

Validity of Findings
Novelty: The study presents innovative findings on miR-146a’s modulation by SA, which is a unique contribution. However:
The novelty is somewhat undermined by the lack of behavioral assays to correlate molecular changes with functional outcomes. For example, nerve conduction studies or pain sensitivity tests would directly link SA treatment to DPN symptom alleviation.

Statistical Rigor: The statistical analyses are appropriate but limited by the small sample size (n=5 per group). Larger sample sizes or additional replicates would increase the reliability of the findings.

Additional comments

Abstract
Comment: The abstract briefly summarizes the findings but lacks clear articulation of the problem being addressed or the hypothesis tested.
Suggestion: Begin the abstract with a concise statement of the clinical or scientific problem (e.g., "Diabetic neuropathy (DN) remains a significant complication of diabetes with limited effective therapeutic options.") followed by the study's objective and the rationale for focusing on sodium alginate (SA).
Elaboration Needed: Include key quantitative results (e.g., percentage improvement in oxidative stress markers) to highlight the study's impact.

Introduction
Comment: While the introduction provides a good overview of DN, it does not sufficiently justify the choice of SA. The relevance of miR-146a in DN pathology is also mentioned but not explored in depth.
Suggestion: Expand on why SA was chosen as the therapeutic agent, citing its specific properties (e.g., antioxidant, anti-inflammatory) and any previous studies that support its potential efficacy in DN.
Elaboration Needed: Include a brief discussion of miR-146a’s role in inflammatory and oxidative stress pathways and how its modulation could address DN pathology.
Materials and Methods

Animal Model:


Comment: The use of a single dose of STZ and the lack of detailed monitoring for diabetic induction limits the replicability of the model.
Suggestion: Provide information on the exact criteria for confirming diabetes induction (e.g., glucose thresholds) and justify why a single dose of 50 mg/kg was chosen.
Elaboration Needed: Specify the timeline for SA administration relative to STZ injection and include details about its preparation and route of administration.

Biochemical Analyses:
Comment: The methodology for assessing oxidative stress markers is adequately described but lacks validation details for the assays used.
Suggestion: Mention whether standard calibration curves were prepared for each marker and include details about assay sensitivity and specificity.

miR-146a Analysis:
Comment: The process for miRNA extraction and quantification is not sufficiently detailed.
Suggestion: Include specifics on the RNA extraction kit, primers used for miR-146a quantification, and normalization controls (e.g., housekeeping miRNA or U6).

Results
Biochemical Markers:
Comment: The improvement in oxidative stress markers (e.g., SOD, CAT) is presented without statistical correlations to the functional parameters.
Suggestion: Add correlation analysis between oxidative stress markers and miR-146a levels to strengthen the mechanistic link.

miR-146a Regulation:


Comment: The findings regarding miR-146a expression are promising but lack insight into downstream targets or pathways.
Suggestion: Include a discussion on whether miR-146a modulation could affect key inflammatory mediators, such as NF-κB or COX-2.

Figures:
Comment: Figures illustrating biochemical and molecular results lack clear legends, making it difficult for readers to interpret the data.
Suggestion: Add detailed figure legends explaining group comparisons and statistical significance markers (e.g., p values).

Discussion
Comment: The discussion reasonably interprets the findings but does not adequately integrate them into the broader context of DN research.
Suggestion: Compare the efficacy of SA to other known therapeutic agents (e.g., alpha-lipoic acid, curcumin) and discuss its potential as a stand-alone or adjunct therapy.

Elaboration Needed: Include a detailed explanation of how miR-146a modulation contributes to the observed improvements in oxidative stress and inflammation.
Comment: The discussion does not address limitations in the study design.
Suggestion: Acknowledge the absence of behavioral assessments (e.g., pain sensitivity, motor function) and propose their inclusion in future studies.

Figures and Tables
Comment: The figures are informative but visually crowded. Bar graphs and line charts need consistent labeling and color coding for clarity.
Suggestion: Simplify the layout by using separate panels for different comparisons and ensure that axis labels and units are visible.

Conclusion
Comment: The conclusion is overly general and does not propose specific future directions.
Suggestion: Summarize the key findings quantitatively and suggest future studies to validate SA's translational potential in larger animal models or clinical trials.

Reviewer 4 ·

Basic reporting

no comment

Experimental design

no comment

Validity of the findings

no comment

Additional comments

In this study the authors studied the Mitigative effect of sodium alginate on streptozotocin (STZ)-induced diabetic neuropathy through regulation of redox status and miR-146a in the rat sciatic nerve. The manuscript is reasonably well written and presented but the novelty is missing. However, there are a few major concerns that should be addressed are shown below:

Authors are suggested to add latest references in line 44-46.
Authors has to correct line number 119……xed in 4F1G solution for pathological examinations.
Authors has to correct line number 180………..fixed in 4F1G, and then they
Authors are suggested to add latest references in line number 319
Authors are suggested to add latest references in line number 314-317
The authors are suggested to add the latest references in the discussion sections.
In line number 53-55 authors suggested to add latest reference about mirna146a.
Figure legend 3 ……….nsPÃ0.05 (non-signiûcant),
Figure 9….authors need to provide triplicate image of each group.
Authors need to provide more information about SA role as an antioxidant in different diseases.
The discussion part has to be revised, and the focus should be more on finding support by latest references.
Transmission electron microscopy…need to provide the reference.
Collection of tissue samples…need to provide the reference.
Authors need to provide the raw file of QPCR from RNA concentration to QPCR analysis file.
Authors are suggested to perform the Western blot, IHC data of some of the proteins.
Authors are suggested to perform behavioral and functional studies in STZ animals’ vs control animals.

---

## Round 0.2 · accepted · Accept

The majority of reviewers accpeted this version for publication.

Reviewer 1 ·

Basic reporting

There are no further comments

Experimental design

There are no further comments

Validity of the findings

There are no further comments

Additional comments

There are no further comments

Reviewer 2 ·

Basic reporting

No comment

Experimental design

No comment

Validity of the findings

No comment

Additional comments

No comment

Reviewer 4 ·

Basic reporting

no comment

Experimental design

no comment

Validity of the findings

no comment